# Uncertainty Quantification in Computer-Aided Diagnosis: Make Your Model say "I don't know" for Ambiguous Cases

**Max-Heinrich Laves**                                      LAVES@IMES.UNI-HANNOVER.DE
**Sontje Ihler**                                            IHLER@IMES.UNI-HANNOVER.DE
**Tobias Ortmaier**                                         ORTMAIER@IMES.UNI-HANNOVER.DE
*Institute of Mechatronic Systems, Leibniz Universitt Hannover, Germany*

## Abstract

We evaluate two different methods for the integration of prediction uncertainty into diagnostic image classifiers to increase patient safety in deep learning[1]. In the first method, Monte Carlo sampling is applied with dropout at test time to get a posterior distribution of the class labels (Bayesian ResNet). The second method extends ResNet to a probabilistic approach by predicting the parameters of the posterior distribution and sampling the final result from it (Variational ResNet). The variance of the posterior is used as metric for uncertainty. Both methods are trained on a data set of optical coherence tomography scans showing four different retinal conditions. Our results shown that cases in which the classifier predicts incorrectly correlate with a higher uncertainty. Mean uncertainty of incorrectly diagnosed cases was between 4.6 and 8.1 times higher than mean uncertainty of correctly diagnosed cases. Modeling of the prediction uncertainty in computer-aided diagnosis with deep learning yields more reliable results and is anticipated to increase patient safety.

**Keywords:** bayesian approximation, variational inference, optical coherence tomography

## 1. Introduction

In recent years, deep learning methods have received significant attention for computer-aided diagnosis in a variety of fields of medical imaging. They outperform former methods in capability and accuracy. However, deep models trained for diagnosis of specific cases currently lack the ability to say "I don't know" for ambiguous or unknown cases. Standard models do not provide prediction uncertainty, which is indispensable for the acceptance of deep learning in medical practice.

Existing approaches for uncertainty estimation in deep learning try to approximate a Bayesian neural network (BNN), where distributions are placed over the weights (Gal and Ghahramani, 2016). BNNs provide the mathematical tool to model uncertainty, but are usually associated with limiting computational cost. Gal et al. have shown, that the use of Monte Carlo sampling with dropout at training and test time acts as approximation to a Bayesian model without aggravating the complexity or the performance of the model (Gal and Ghahramani, 2016; Kendall et al., 2015). This idea has attracted attention in medical segmentation (Roy et al., 2019; Laves et al., 2019). Another method for uncertainty estimation is a variational inference approach, where a deep model is trained to learn the

---

1. Code is available at https://github.com/mlaves/uncertainty-midl

parameters of a probability distribution from which the final prediction is sampled (Kingma and Welling, 2013). Variational inference has been used recently for uncertainty estimation in deformable registration of brain MRI (Dalca et al., 2019).

In this work, the aforementioned approaches for uncertainty estimation are integrated into diagnostic classifiers and compared in order to increase patient safety and acceptance for deep models in medical imaging.

## 2. Methods

In the following, we will briefly revise the Bayesian and variational inference approach for uncertainty estimation. Given a set of training images $\boldsymbol{X}$ with corresponding labels $\boldsymbol{Y}$ from medical experts, we try to find a probabilistic function $f_\theta : x \to y$ yielding the most likely label prediction $\hat{y}$ of a test image $x$ with probability

$$p(\hat{y}|x, \boldsymbol{X}, \boldsymbol{Y}) = \int p(\hat{y}|x, \theta)p(\theta|\boldsymbol{X}, \boldsymbol{Y})\mathrm{d}\theta \ , \tag{1}$$

with parameters $\theta$ of the deep model $f$. The posterior distribution in (1) is generally intractable and therefore, the integral can be approximated by summing Monte Carlo samples obtained from $f_\theta$ with dropout at test time (Gal and Ghahramani, 2016). The mean of these samples is used as label prediction $\hat{y}$ and the variance is interpreted as the uncertainty of the prediction. We train the ResNet-18 image classifier on a dataset of 84,484 optical coherence tomographies showing four different retinal conditions (He et al., 2016; Kermany et al., 2018). Dropout with $p = 0.5$ is added before the last fully connected layer (referred to as *bayesian1*) and before every building block of ResNet-18 (referred to as *bayesian2*), creating a bayesian classifier. In Monte Carlo experiments, 100 forward passes are performed to get an approximation of the posterior distribution of the class labels.

In the variational inference approach, we assume a normal distribution for the posterior and replace the last fully-connected layer of ResNet with two fully-connected layers predicting the parameters $\boldsymbol{\mu}$ and $\boldsymbol{\sigma^2}$ of the posterior distribution (referred to as *variational*). The final prediction is sampled from this distribution using the reparameterization trick $\hat{\boldsymbol{y}} = \boldsymbol{\mu} + \boldsymbol{\sigma}\boldsymbol{\varepsilon}$ with $\boldsymbol{\varepsilon} \sim \mathcal{N}(\mathbf{0}, \mathbf{I})$. As proposed in (Kingma and Welling, 2013), we additionally try to bring the estimated posterior closer to a standard normal distribution by adding the Kullback-Leibler divergence (KLD) to the overall loss function. In this case, the KLD can be solved analytically as $\mathcal{L}_{\mathrm{KLD}}(\boldsymbol{\mu}, \boldsymbol{\sigma}^2) = -\frac{1}{2}\sum_{j=1}^{N}\left(1 + \log(\sigma_j^2) - \mu_j^2 - \sigma_j^2\right)$.

After training the three approaches and a *baseline* ResNet-18 for comparison, we investigate the uncertainties for all true and false predictions for images from the test set.

## 3. Results

Fig. 1 show boxplots (top row) and relative frequencies (bottom row) for uncertainties of correctly (true) and incorrectly (false) predicted cases from the test set. The results shown that cases in which the network predicts incorrectly correlate with a higher uncertainty. Mean uncertainty of incorrectly diagnosed cases was 4.6 (variational), 6.0 (bayesian2) and 8.7 (bayesian1) times higher than mean uncertainty of correctly diagnosed cases. Test set accuracies compared to a baseline ResNet-18 are listed in Tab. 1.

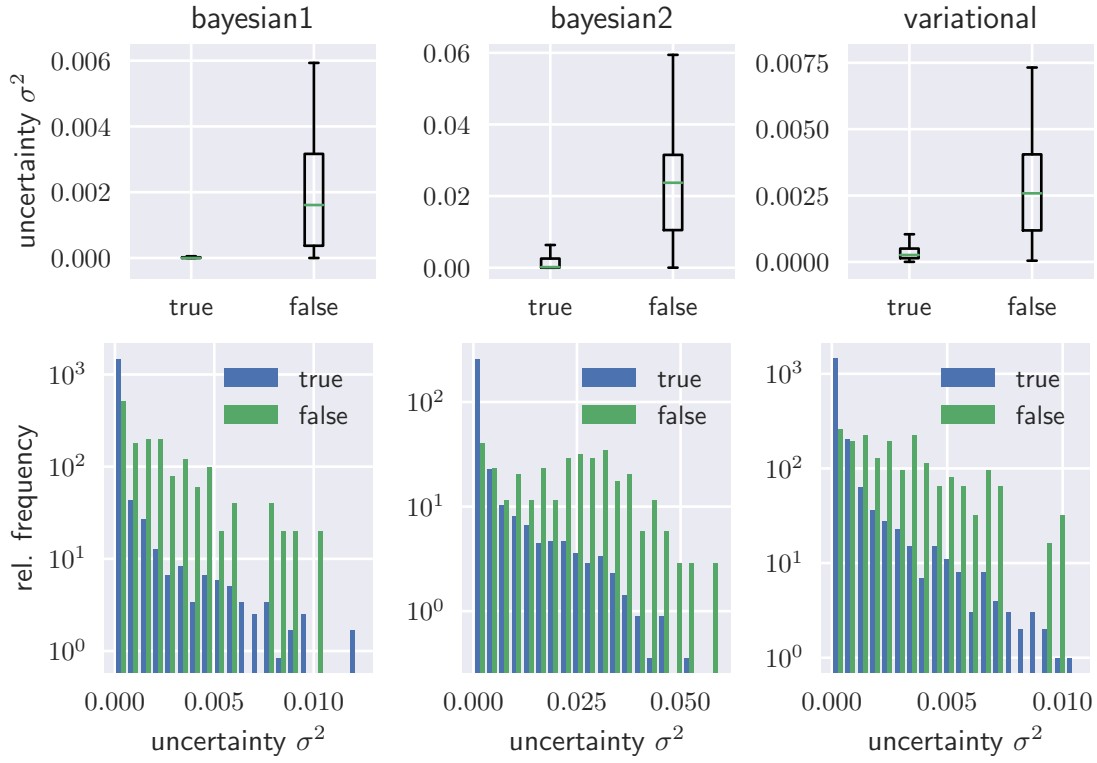

Figure 1: Results from different approaches for uncertainty estimation of correctly (true) and incorrectly (false) diagnosed OCT scans. All methods yield a higher uncertainty for incorrect predictions.

|  | baseline | bayesian1 | bayesian2 | variational |
|---|---|---|---|---|
| precision | 0.96 | 0.96 | 0.93 | 0.94 |
| recall | 0.95 | 0.96 | 0.93 | 0.94 |
| F1 score | 0.95 | 0.96 | 0.93 | 0.94 |

Table 1: *Bayesian1* does not affect test set accuracy. Many dropout layers in *bayesian2* and the noise introduced by the reparameterization trick in *variational* seem to have a negative effect.

## 4. Conclusion

Modeling of the prediction uncertainty in computer-aided diagnosis with deep learning yields more reliable results and is therefore anticipated to increase patient safety. This can help to transfer such systems into clinical routine and to increase the acceptance of physicians and patients for machine learning in diagnosis. In future work, the uncertainties can be used to further increase classification accuracy.

## Acknowledgments

This research has received funding from the European Union as being part of the EFRE OPhonLas project.

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
