# OpenReview forum: "Uncertainty Quantification in Computer-Aided Diagnosis: Make Your Model say “I don’t know“ for Ambiguous Cases"
_MIDL.io/2019/Conference/Abstract — MIDL Abstract 2019_

### Official Review · AnonReviewer1 · 2019-04-30
**Interesting abstract**

**Rating:** 3
**Confidence:** 3

**Review:**

This paper investigates the use of uncertainty estimators is deep networks. This is an interesting work and I vote for accepting it.
The results in Table 1 actually show that bayesian2 and variational estimators actually drop the results a bit. This is in contrast with what the authors mentioned in the caption.

---

### Official Review · AnonReviewer2 · 2019-05-01
**.**

**Rating:** 2
**Confidence:** 3

**Review:**

The paper applies standard theory on uncertainty estimation in neural nets to the problem of medical image classification. While this effort is in the right direction, the results don't show significant insight into the medical domain.

---

### Decision · Program_Chairs · 2019-05-06
**Acceptance Decision**

Accept